# Impact of the Processing-Induced Orientation of Hexagonal Boron Nitride and Graphite on the Thermal Conductivity of Polyethylene Composites

**DOI:** 10.3390/polym15163426

**Published:** 2023-08-16

**Authors:** Mehamed Ali, Patrik Sobolciak, Igor Krupa, Ahmed Abdala

**Affiliations:** 1Chemical Engineering Program, Texas A&M University at Qatar, Doha P.O. Box 23874, Qatar; 2Center of Advanced Materials, Qatar University, Doha P.O. Box 2713, Qatar

**Keywords:** thermal conductivity, polyethylene composites, hBN, graphite, filler alignment

## Abstract

Emergent heat transfer and thermal management applications require polymer composites with enhanced thermal conductivity (κ). Composites filled with non-spherical fillers, such as hexagonal boron nitride (hBN) and Graphite (Gr), suffer from processing-induced filler orientations, resulting in anisotropic κ, commonly low in the through-plane direction. Here, the effects of extrusion and compression molding-induced orientations on κ of hBN- and Gr-filled polyethylene composites were investigated. The effect of extrusion on the hBN orientation was studied using dies of various shapes. The shaped extrudates exhibited hBN orientations parallel to the extrusion flow direction, which prompted additional hBN orientation during compression molding. κ of the composites produced with shaped extrudates varied from 0.95 to 1.67 W m^−1^ K^−1^. Pelletizing and crushing the extrudates improved κ, by exploiting and eliminating the effect of extrusion-induced hBN orientations. Gr-filled composites showed better κ than hBN composites due to the higher intrinsic conductivity and bigger particle sizes. A maximum κ of 5.1 and 11.8 W m^−1^ K^−1^ was achieved in composites with oriented hBN and Gr through a thin rectangular die and stacking the sheets to fabricate composites with highly oriented fillers.

## 1. Introduction

Polymers and their composites are widely used in industrial and daily life applications due to their excellent attributes, such as lightweight, corrosive resistance, electrical resistivity, ease of fabrication, and low cost [1,2]. One polymer composite application is in the thermal dissipation management of electronic devices that usually requires high thermal conductivity [3]. Moreover, polyethylene (PE) is used in the insulation of power cable, which requires low electrical conductivity and high thermal conductivity [4,5,6]. However, the thermal conductivity (κ) of polymers is very low, in the range of 0.1–0.5 W m^−1^ K^−1^ [7]. Thus, enhancing κ of polymers is a preliminary requirement for thermal management applications. This is typically done by the addition of thermally conductive inorganic fillers, such as carbon fiber (CF) [8], carbon nanotubes (CNT) [9], graphene [10], Gr [11], and hBN [12].

Non-spherical fillers, including hBN, CF, CNT, and Gr, are usually desirably, or undesirably, orientated during processing, such as during extrusion [13], injecting molding [14,15], and compression molding [16,17]. Figure 1a,b show the two extremes, perpendicular and parallel filler orientations. Due to the anisotropic properties of such fillers, their orientations usually lead to composites of anisotropic properties [18,19,20]. For instance, the through-plane and in-plane κ of hBN are ~2–30 W m^−1^ K^−1^ and ~300–600 W m^−1^ K^−1^ [7,21]. Consequently, their composites display anisotropic κ. Ahn et al. [22] reported that κ of poly(vinyl butyral) composites improved by approximately a factor of 20 when the hBN particles were oriented parallel to the heat flow (in-plane) compared to when they were aligned perpendicular to the heat flow (through-plane) [22]. Ghose et al. [13] used a laboratory mixing extruder to align CF, multi-walled CNTs, and expanded graphite in UltemTM 1000 matrix. The fillers aligned in the flow direction, leading to an anisotropic thermal conductivity reaching a maximum of 6.78 W m^−1^ K^−1^ in 40 wt.% expanded graphite/ UltemTM 1000 composite. Meanwhile, using injection molding, Zhang et al. [15] aligned alumina fiber in high-density polyethylene (HDPE). They found that 15–37% and 63–85% of the fibers were oriented perpendicular and parallel to the injection molding direction, respectively. Equivalently, Yoo et al. aligned graphite and CF hybrid in polyamide polymer using injection molding, leading to through-plane and in-plane κ of 2.03 and 5.09 W m^−1^ K^−1^ at 60 wt.% loading [23]. Unlike injection molding, compression molding leads to composites with fillers aligned perpendicular to the compression molding direction and parallel to the in-plane direction [16,17].

Several techniques have been devised to align hBN preferentially and Gr in the direction of heat flow, including magnetic field [21,24], electrical field [25,26], extrusion [13], squeeze casting [27], hot-pressing [17,28], tape-casting [29,30], and multilayer stacking and annealing [31]. However, detailed studies are lacking on the flow-induced orientations, which irresistibly occur during the composite fabrication processes, i.e., extrusion and compression molding. Some studies have shown that fibers align parallel, in the edge (skin layer), and perpendicular, in the center (core layer), to the injection molding flow direction, a phenomenon known as skin layer and core layer shell structures, as shown in Figure 1c [32,33,34]. The same phenomenon is expected for composites filled with hBN and Gr fillers during compression molding.

This study investigated the effect of extrusion and compression molding processes on the hBN and Gr orientations and, after that, on the thermal conductivity of polyethylene composites. The skin layer and core layer structures were examined at different processing parameters. Composites with random, parallel, and perpendicular hBN orientations were prepared and examined for comparison purposes. To correlate the composite properties with their morphologies and crystal structure, these composites were analyzed using SEM and XRD. In addition, several models were used to predict the thermal conductivities of these composites.

## 2. Materials and Methods

### 2.1. Materials

Hexagonal boron nitride (hBN) powder with 35 µm average particle sizes and 99.0% purity was acquired from Asbury Carbon, Asbury, NJ, USA. Natural graphite flakes (grade 3061) with 99% carbon concentration and + 50 mesh particle size (>80%) were obtained from Sigma Aldrich, St. Louis, MO, USA. Linear low-density polyethylene (LLDPE) powder with density of 0.916 g/cm^3^ and melting point of ~122 °C (Lotrene Q2018) was kindly supplied by Qatar Petrochemical Company (QAPCO), Doha, Qatar. HDPE pellets with density of 0.938 g/cm^3^ and melting point of ~134 °C (Lotrene Q3802) were kindly supplied by Qatar Chemical Company Ltd. (Q-Chem), Doha, Qatar.

### 2.2. Composites Preparation

The composites were prepared by Xplore Twin Screw 15 mL Micro-Compounder (Xplore MC 15-HT, Sittard, The Netherlands). In each batch, 9 g of the composites (e.g., hBN powder and HDPE), were premixed and fed to the extruder operating at 210 °C, 100 rpm rotation rate, and 10 min residence time. The extrudates were shaped using several extruder dies to form circular (C), triangular (T), square (Sq), and rectangular (R) solid tubes or strands, as shown in Figure 2a. Afterward, the extrudates were pressed into sheets using Craver Bench Top Heated Manual Press (Model 4386) at a temperature of 180 °C, a pressure of ~10 bar, and 5 min residence time. The C-shaped extrudates were powdered (Figure 2a) using a simple kitchen grinder and pressed into sheets of various thicknesses to investigate the effect of compression molding on the filler orientations.

The impact of compression molding was studied by pressing the C-shaped extrudates into sheets of 0.5- and 2-mm thicknesses. In addition, the effect of compression pressure and composite sheet thickness was further investigated by preparing composites using a custom-designed mold press (CDP). To study the impact of extrusion on the filler orientations, the composites made from different extrudates were compared to each other, and the composite made from the sample collected directly from the extruder barrel (AE sample).

The R-shaped extrudates were pressed perpendicular (P) and random (r) relative to their longest side. For instance, R1P and R1r mean R1 is pressed perpendicular and random to the 5 mm side, respectively. Please note that the random pressing of the R-shaped strands was not controlled because the mold thickness (2 mm) was smaller than the width of the strand. To further investigate the effect of extrusion, the C, T, and Sq-shaped strands were pelletized into pellets (Pe) of 0.5–1 mm thicknesses to ensure that when pressed, the pellets lie down facing the strand’s cross-section. Finally, composites with highly oriented fillers were synthesized through a simple method, in which the R3 strands were stacked to form bars of 1 × 1 × 4 cm dimensions, as shown in Figure 2c. A multilayer sheet stacking method proposed by Yang’s group was adopted and modified [31]. Several hBN/LLDPE composite sheets of 0.3–0.5 mm thicknesses were stacked and rotated 90°. LLDPE composites were chosen over HDPE for their lower viscosity and ease of pressing.

### 2.3. Characterization

The structure and morphology of hBN, Gr, and their LLDPE and HDPE composites were analyzed using VERSA 3D (FEI) SEM. XRD analysis was performed in the 2θ range of 5–65° at a scan rate of 0.05°/min and a step size of 0.01° using Ultima IV X-RAY Diffractometer (Rigaku). TGA was conducted using Discovery TGA (TA Instruments, New Castle, DE, USA) under a nitrogen atmosphere at a heating rate of 10 °C/min and a temperature range of 25–700 °C. The thermal conductivity (κ) was determined by measuring the thermal diffusivity obtained using LFA 500 laser flash (Linseis, Selb, Germany) at room temperature using a ½-inch disk with a thickness of 0.5–1.5 mm.

## 3. Results

Due to their plate-like geometry, hBN particles tend to align during processing, i.e., extrusion and compression molding. The SEM images of the hBN and hBN/HPDE composites are shown in Figure 3. The hBN particles, shown in Figure 3a, were multilayered sheets of sizes in the range of 10 to 50 µm. The cross-sectional surface of pure HDPE was smooth due to the absence of fillers, as shown in Figure 3b. The C and R3 strands displayed hBN orientations parallel to their outer surface and the extrusion flow direction, as shown in Figure 3c,d. Therefore, to only study the compression molding-induced hBN orientations, the C strands were powdered before pressing to eliminate the extrusion-induced hBN orientations.

### 3.1. Compression Molding Induced hBN Orientations

Figure 4 shows κ of the powdered composites loaded with 30 vol.% hBN at various sheet thicknesses. Generally, κ demonstrated strong dependence on thickness, and hence compression molding. In principle, κ increased from ~1 W m^−1^ K^−1^ at 0.5 mm to ~2.5 W m^−1^ K^−1^ at 3 mm. To correlate κ of these composites with morphology, these composites were analyzed using SEM, as shown in Figure 3g–i. In the skin layer, the hBN particles were aligned perpendicular to the compression molding direction, while in the core layer, they were randomly aligned. These hBN orientations were similar to previously discussed skin and core layer structures, refer to Figure 1b. These hBN orientations were induced during the compression molding (force) was high at the skin layer forcing the hBN particles to orient perpendicular to the compression direction.

Conversely, the compression force on the core layer was relatively weak, leading to parallel or random hBN orientations. The skin layer thickness hardly varied with the sheet thickness and was found to fall in the range of 0.08–0.15 mm, supporting the above argument. Meanwhile, the core layer thickness increased with the composite thickness. Thus, the increase of only the core layer thickness improved κ with the sheet thickness. To theoretically evaluate the effect of thickness on κ, the composites κ at each thickness was calculated using the following model [1,35]:(1)$$R=R_{skin}+R_{core}$$
(2)$$R=\frac{t}{\kappa }$$
(3)$$R_{skin}=\frac{t_{skin}}{\kappa_{skin}}$$
(4)$$R_{core}=\frac{t_{core}}{\kappa_{core}}$$
(5)$$\kappa =\frac{t}{\frac{t_{skin}}{\kappa_{skin}}+\frac{t_{core}}{\kappa_{core}}}$$
where *R*, *R_skin_*, and *R_core_* are the overall, skin layer, and core layer thermal resistances, respectively, and the resistance is defined by Equation (2), in which ‘*t*’ (mm) is the layer thickness. As illustrated in Figure 4, the model reasonably predicted the experimental κ. In this model, *t_skin_*, *κ_skin_*, and *κ_core_* were constants, as shown in Figure 4. The model-predicted skin layer thickness of 0.115 mm agrees well with the skin layers’ thickness observed by SEM analyses. As expected, the skin layer exhibited a low κ of 0.568 W m^−1^ K^−1^, meanwhile, the core layer showed a high κ of 3.59 W m^−1^ K^−1^. Therefore, the core layer thickness increases with the composite sheet thickness, leading to enhancement of κ.

Figure 5a shows κ of hBN/HDPE composites prepared by pressing the C-shaped strands into 0.5 mm (C-0.5 mm) and 2 mm (C-2 mm) thick sheets. The C-2 mm composites exhibited increased κ. For example, at ~40 vol.%, κ of the C-0.5 mm and C-2 mm composites were 1.43 and 2.72 W m^−^^1^ K^−^^1^, respectively. As discussed earlier, the hBN particles of the C strand were oriented parallel to its outer surface (Figure 3c). Consequently, this extrusion-induced orientation dictated the hBN orientations in the press-molded composite sheets, resulting in a sheet with regions of different hBN orientations relative to the heat flow direction. In the skin layer, the hBN orientation was perpendicular, whereas, in the core layer, there were several hBN orientations. For example, the C-2 mm composites exhibited three different hBN orientations depending on the distance from the outer surface of the composite sheet, as shown in Figure 3j–l. In the skin layer, the C-2 mm composites displayed perpendicular hBN orientation. While, in the core layer, they displayed various orientations, i.e., the hBN particles were inclined with an angle greater than 0° and less or equal to 90°, where 0° and 90° correspond to the perpendicular and parallel hBN orientations, respectively.

On the other hand, the C-0.5 mm composites displayed mainly perpendicular hBN orientations, as shown in Figure 3m. Figure 2b illustrates the effect of compression molding on the hBN orientation schematically. The compression force was dominant in the thin sheet (C-0.5 mm), resulting in a composite with mostly perpendicular hBN orientation. Conversely, in the 2 mm thick sheet (C-2 mm), the effect of compression force was only pronounced at the skin layer. Hence, moving toward the core layer, the force weakens, resulting in hBN particles of different orientations depending on the distance from the skin layer. The through-plane κ of hBN is in the range of 3–30 W m^−1^ K^−1^, thus, composites with perpendicular hBN orientation exhibit low κ, i.e., κ displayed by the C-0.5 mm composites.

To further investigate the skin and core layer structures, the skin layer of the C-2 mm composites was removed by simple polishing at an interval of ~0.25 mm, as illustrated in Figure 5b. Figure 5b shows κ of the 30 and 39 vol.% hBN/HDPE and 39 vol.% hBN/LLDPE composites at each removed skin layer thickness. As expected, κ was increased as the skin layer was removed. Notably, when the removed skin layer thickness increased from 0 to 0.74 mm, κ of the 39 vol.% hBN/HDPE increased from 2.72 to 3.44 W m^−1^ K^−1^. However, further removal of the skin layer resulted in a negligible increase in κ.

In contrast, κ of hBN/LLDPE composite increased for a larger range of removed skin layer thickness, i.e., 0–1.51 mm, due to the viscosity difference between the two composite types. For instance, the HDPE composites, having higher melt viscosity, were expected to be more resistant to compression molding-induced hBN orientations. As a result, the hBN/HDPE composites displayed a smaller skin layer thickness than the hBN/LLDPE composites. It is worth noting that the removed skin layer thicknesses of the 30 vol.% and 39 vol.% HDPE composites, after which there was no significant change in κ were ~0.5 and 0.7 mm, respectively. These skin layer thicknesses were approximately equal to the sheet thickness of the C-0.5 mm composites, justifying their perpendicular hBN orientation and decreased κ. It is also worth noting that the thickness of the skin layer of the C-2 mm composites was much bigger than their powdered equivalents, leading to their comparatively lower κ (as will be shown in the following section).

### 3.2. Extrusion-Induced hBN Orientations

Figure 6a shows κ of 30 vol.% hBN/HDPE composites produced in various shapes and pressed into 2 mm thick sheets. κ of the composites varied from 0.945 W m^−1^K^−1^ in the R3P composite to 1.74 W m^−1^ K^−1^ in the AE composite. κ of the AE composite was higher than the shaped composites, indicating that some unfavorable hBN orientations were taking place during the shaping step, i.e., shaping the extrudates. Among the shaped composites, κ of the R1P, R2P, and R3P composites were the lowest, and R2r and Sq composites were the highest. On the other hand, C (C-2 mm), T, and R1r composites manifested similar and intermediate κ. The SEM images of the C and R3 strands displayed parallel hBN orientation at the skin layer; this hBN orientation was also parallel to the extrusion flow direction, as shown in Figure 3c,d, respectively. The other strands, not shown in Figure 3, manifested similar hBN orientations at the skin layer. The core layer’s C, T, and R strands also showed parallel hBN orientation relative to the extrusion flow direction. For example, Figure 3e shows the core layer of the C strand, which exhibited hBN orientation parallel to the extrusion flow direction. However, the Sq strand, shown in Figure 3f, displayed relatively random hBN orientation at the core layer. This was ascribed to the comparatively bigger pores of the Sq strand, which was caused by its bigger cross-sectional area. For instance, the cross-sectional area of the Sq strands was 25 mm^2^, compared to 7, 9, 10, 20, and 10 mm^2^ of the C, T, R1, R2, and R3 strands, respectively, giving the hBN particles more orientational freedom. Thus, their composite was comparatively resistant to extrusion and compression molding-induced hBN orientations, justifying their relatively higher κ. Despite having well-arranged hBN particles, the symmetry of the C strands led to composites with slightly unfavorable hBN orientations. For example, it has been shown earlier in Figure 3j–l that the C-2 mm composites displayed skin and core layer hBN orientations, in which the skin layer was detrimental to κ. The similarity of κ of the T and C composites was attributed to their cross-sectional symmetry and approximately equal area. For instance, it was inferred that the equilateral nature and the small sides of the cross-section of the T strand have led to composites with hBN orientations similar to the C composites.

Conversely, the R strands were asymmetrical; hence, their resultant composites displayed (with considerable dependence on the compression molding direction). Remarkably, the R strands exhibited highly arranged hBN particles parallel to their long side, and when pressed opposite to this side (R1P, R2P, and R3P), their composites displayed low κ. However, when pressed randomly (R1r and R2r), their composites increased in κ. κ of R1 and R2 composites was increased from 1.24 to 1.46 W m^−1^ K^−1^ and 0.97 to 1.61 W m^−1^ K^−1^, respectively, when their strands were pressed randomly instead of perpendicular. The R2 strands accredited to their long side and more arranged hBN particles were more prone to extrusion and compression molding induced hBN orientations, leading to the most significant change in κ when the pressing direction was altered.

Motivated by the improvement in κ of the R1r and R2r composites, composites with high κ in the through-plane direction were fabricated by stacking the R3 strands to form rectangular bars of 1 × 1 cm sides, as shown in Figure 2c. Subsequently, κ was measured parallel and perpendicular to the longer side of the original strands and named R3|| κ and R3⏊ κ, respectively. Figure 6b shows the R3|| and R3⏊ κ values of 30 vol.% hBN/HPDE composites at different hBN concentrations. At all hBN loadings, R3|| κ was substantially higher than R3⏊ κ. Remarkably, at 10 and 30 vol.% hBN contents, R3|| κ were 2.3 and 5.15 W m^−1^ K^−1^, respectively. Meanwhile, R3⏊ κ were ~0.87 and 1.44 W m^−1^ K^−1^, respectively. The morphology of the R3 strand and the R3 composites, shown in Figure 3d,n,o, respectively, exhibited highly aligned hBN particles. The R3 strand displayed parallel hBN orientation relative to its more extended side surface, and subsequently, the R3 composites exhibited perpendicular and parallel hBN orientation in the R3⏊ and R3|| directions, respectively. The remarkably high R3|| κ was justified by the parallel hBN orientation in the R3|| direction. On the contrary, despite the perpendicular hBN orientation in the R3⏊ direction, R3⏊ κ was relatively high compared to composites with similar hBN orientation. This suggested that the hBN alignment was not perfectly parallel or perpendicular in the R3 composites.

The composites were analyzed by XRD to correlate these findings, as shown in Figure 7. As expected, the (002) peak, which intensifies in the perpendicular hBN orientation [31], diminished or almost disappeared in the R3|| direction (Figure 7a), supporting the SEM and κ results. Figure 7b, which is the normalized version of Figure 7a, shows the (100) peak, which intensifies in the parallel hBN orientation [31], surpassing the (002) peak in the R3|| direction. For instance, the ratios of the (100) peak to the intensity of (002) peak were 0.0025 and 1.83 in the R3⏊ and R3|| directions, respectively.

To further investigate this point the R3 composites were compared with composites prepared by modifying Yang’s method of sheet stacking and rotation [31]. Figure 6c shows the κ of the 22.5 vol.% hBN/LLDPE composites prepared by the R3 method (R3⏊ and R3||), modifying Yang’s method (Stacked), and by pressing C composites into 0.5 mm (C-0.5 mm) and 2 mm (C-2 mm) sheets. Generally, as was seen in the HDPE-based composites, the hBN/LLDPE composites κ displayed functionality with the hBN orientations. For example, κ varied from 0.56 W m^−1^ K^−1^ in the C-0.5 mm to 3.9 W m^−1^ K^−1^ in the stacked, owing to their perpendicular and parallel hBN orientations, respectively.

Moreover, the R3⏊ κ of 0.96 W m^−1^ K^−1^ was slightly higher than that of the C-2 mm (0.80 W m^−1^ K^−1^), and the 2.93 W m^−1^ K^−1^ for R3|| κ was lower than that of the stacked (3.9 W m^−1^ K^−1^), suggesting that the hBN orientations were not completely perpendicular or parallel in the R3⏊ and R3|| directions, respectively. Owing to the better crystallinity and higher intrinsic κ of the HDPE polymer, R3|| κ of HDPE-22.5% hBN composite was 4.62 W m^−1^ K^−1^, much higher than that of the hBN/LLDPE composite (2.93 W m^−1^ K^−1^). The R3|| κ of the HDPE- hBN composite was even higher than the stacked hBN/LLDPE composite, suggesting that the stacked hBN/HDPE composite would have higher κ. Nonetheless, R3 strands provide a more facile and practical approach than Yang’s. For example, for maximum κ in Yang’s method required pressing the composites into extremely thin sheets (<< 0.5 mm), followed by stacking the sheets. This was strenuous, time-consuming, and not possible at high filler loading, especially in HDPE-based composites.

As was realized previously, the shaping of the extrudates resulted in hBN orientation that, when pressed, was detrimental to κ. Therefore, the shaped extrudates were pelletized into thin pellets to ensure they were pressed perpendicular to their original strands. Figure 6d shows κ of the 30 vol.% hBN/HDPE composites prepared by pelletizing or powdering the shaped extrudates. In addition, Figure 6d compares the effect of hBN orientation on κ by including composites with almost perpendicular and parallel hBN orientation, such as R3–0.5 mm and R3||, respectively. The pelletized composites generally exhibited enhanced (over their shaped extrudates counterparts). In particular, the C-Pe, T-Pe, and Sq-Pe composites demonstrated 53, 59, and 6.2% increase in κ over their shaped strands equivalents, respectively. As anticipated, the κ values of the Sq composites before and after pelletization demonstrated minor variation, which was related to the less hBN orientations of the Sq strands. The powdered composite displayed 75 and 14% increase in κ over the C and C-Pe composites, respectively. The reason for this enhancement in κ was attributed to the random hBN orientation and smaller skin layer of the powdered composites. However, how they outperformed the pelletized composites was not completely understood. Perhaps some pellets were pressed in a direction similar to their original strands, leading to lower κ. Comparing κ as a function of hBN orientations, it can be seen that κ increased from 0.77 W m^−1^ K^−1^ in the R3–0.5 mm composite, with almost perpendicular hBN orientation, to 5.14 W m^−1^ K^−1^ in the R3|| composite, with almost parallel hBN orientation. As expected, composites with random hBN orientation, such as the shaped strands, pelletized, and powdered composites, exhibited intermediate κ. As seen in the hBN/LLDPE composite, the R3⏊ κ was higher than that of the C-0.5 mm composite and approximately equal to that of the C composite. This was ascribed to the imperfect hBN orientation in the stacked R3 composites.

### 3.3. Extrusion and Compression Molding Induced Graphite Orientations

Figure 8a compares κ of 30 vol.% hBN/HPDE and Gr/HDPE composites prepared by pelletizing the Sq and C strands and stacking the R3 strands. The Gr-filled composites demonstrated better κthis was ascribed to the higher intrinsic κ and bigger particle size of the Gr filler, i.e., the average particle size of hBN and Gr were ~35 and 300 µm, respectively. The substantially bigger Gr particles led to lower interfacial thermal resistance because large fillers form fewer filler-polymer interfaces resulting in higher κ [1]. Fan et al. [36] found Gr more efficient in enhancing κ compared to BN, which they assigned to its bigger particle size. Apart from the particle size, κ values of both hBN and Gr-filled composites showed similar trends in the different composite types. This was attributed to their similar plate-like geometry, which expectedly led to composites of similar filler orientations. Notably, κ increased with the improvement in the filler orientations reaching a maximum of 11.8 W m^−1^ K^−1^ in the R3|| Gr/HDPE composite.

To further investigate the effect of compression molding on the filler orientation and after that κ, the C and Sq pellets were pressed into ~1- and 3-mm thick sheets using a custom-designed mold press (CDP) of a little pressure, i.e., its weight. The composites prepared by the CDP showed higher κ than those fabricated by the conventional compression mold (Craver), Figure 8b. In addition, κ of the composites prepared by the CDP was influenced by the sheet thickness of the composite sample. The decrease in κ with the increase in the compression molding force, i.e., high pressure (CDP vs. craver) or small thickness (CDP 1 vs. CDP 3), was directly related to its impact on the filler orientations. The SEM images of the CPD 3 composites showed some pores. Meanwhile, the CDP 1 composites displayed no pores; caused by the poor compression molding. Despite having pores, κ of the CDP 3 composites were high, perhaps, these composites formed filler networks that were more effective in enhancing κ.

## 4. Conclusions

This study investigated the effect of processes, such as extrusion and compression molding, induced filler orientations in hBN and Gr-based polyethylene (HDPE and LLDPE) composites. hBN/HDPE composite strands of different shapes were prepared through extrusion equipped with various dies. The hBN particles orient differently in each strand shape; therefore, pressing these strands prompted additional hBN alignment, leading to composites with slightly lower κ. For example, κ of the C composites were substantially higher when pressed into 2 mm thick sheets, compared to hot-pressing them into 0.5 mm thick sheets—powdering these composites to eliminate the extrusion-induced hBN orientations led to composites with slightly increased κ. κ of 5.14 W m^−1^ K^−1^ was achieved when the extrusion products were preferentially aligned in a thin rectangular strand and subsequently pressed in a direction with the best hBN orientations. This κ was 3 and 2 times κ of the C-2 mm and powdered composites, respectively. A maximum κ of 11.8 W m^−1^ K^−1^ was observed in Gr/HDPE composite with highly arranged Gr particles. Moreover, incorporating of hBN and Gr in the matrix of XLDPE can enhance κ as well as the mechanical properties. Therefore, this investigating the impact of hBN and Gr on the properties of XLDPE will be the subject of a future study.

## Figures and Tables

**Figure 1 polymers-15-03426-f001:**
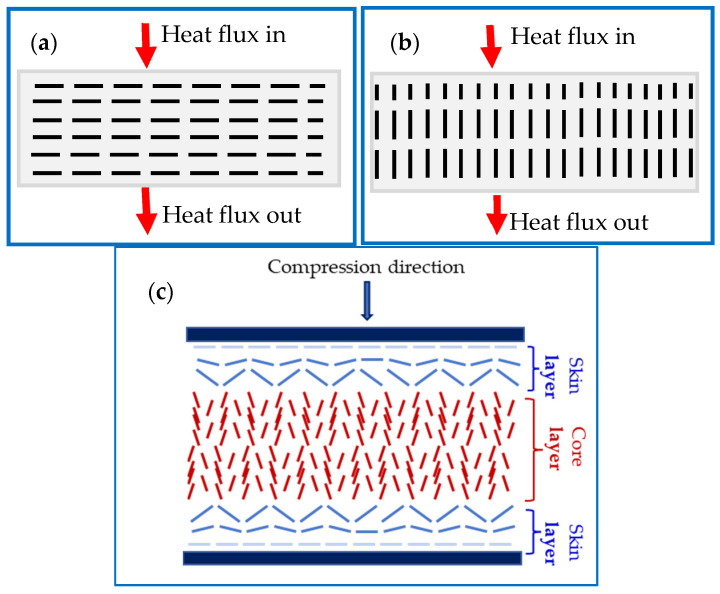
(**a**) Parallel and (**b**) Perpendicular orientation of 2D fillers and (**b**) Skin layer and (**c**) core layer filler distribution induced by compression molding.

**Figure 2 polymers-15-03426-f002:**
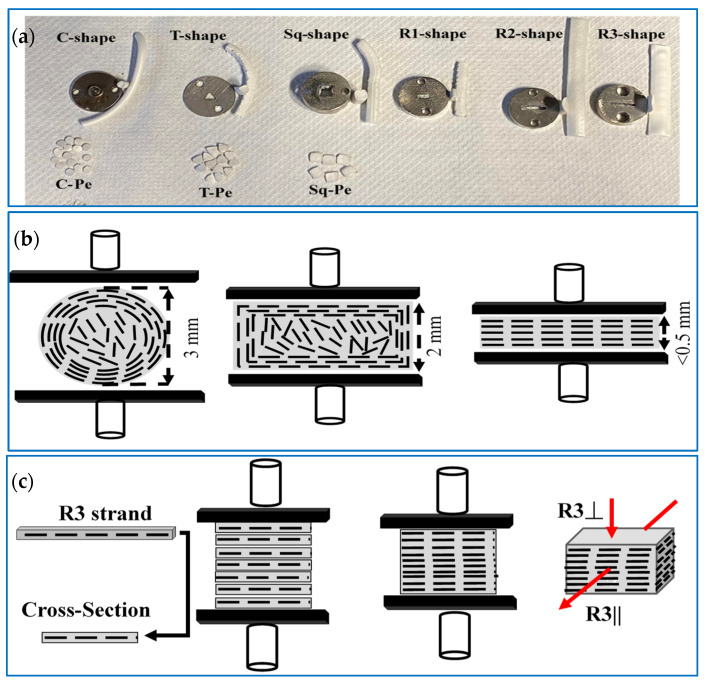
(**a**) Extruder dies and the shape of their corresponding strands and pellets, schematic of the flow-induced hBN orientations during compression molding of (**b**) C-shaped strands, and (**c**) R3-shaped strands. The rectangular strands R1, R2, and R3 have the dimensions of 2 × 5, 2 × 10, and 1 × 10 mm, resepctively.

**Figure 3 polymers-15-03426-f003:**
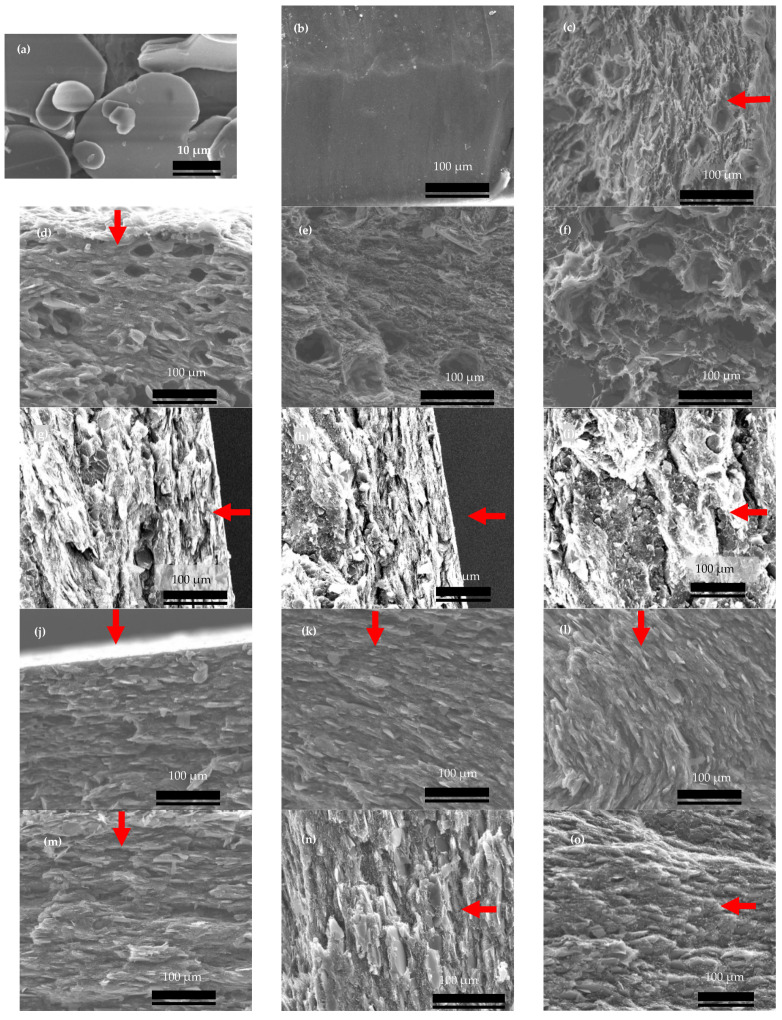
SEM images of (**a**) bulk hBN, (**b**) pure HDPE and HDPE-30% hBN composite samples pressed from strands (**c**–**f**) and powdered samples (**g**–**i**), including (**c**) the skin layer of the C-shaped strand, (**d**) the skin layer of R3-shaped strand, (**e**) the core layer of the C-shaped strand, (**f**) the core layer of the Sq-shaped strand, (**g**) the skin layer of 0.5-mm thick sheet from powder samples, (**h**) the skin layer of 1-mm thick sheet from powder sample, and (**i**) the core layer of 1-mm thick sheet from powder sample, (**j**) skin layer and (**k**,**l**) the core layer of the C-2 mm composite, (**m**) the core layer of the C-0.5 mm composite, (**n**) R3 composites in the orthogonal direction (R3⏊), and (**o**) R3 composites in the parallel direction (R3||). All images were captured from the cross-section, and the red arrows indicate the direction from the outer surface (skin layer) to the center (core layer).

**Figure 4 polymers-15-03426-f004:**
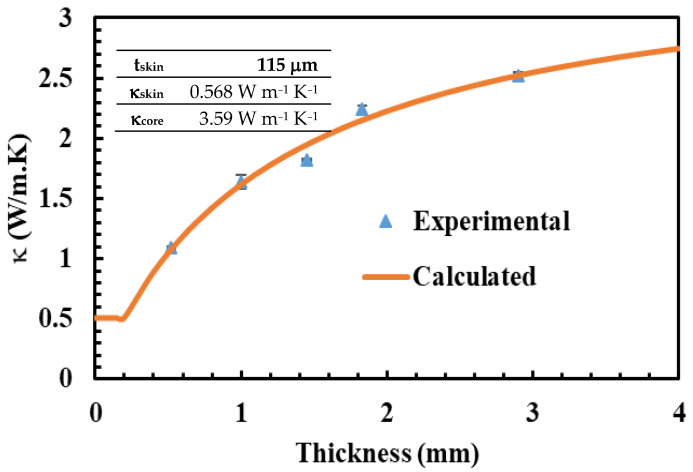
κ of 30 vol.% hBN/HPDE powdered composites at different sheet thicknesses were experimentally measured (symbols) and calculated using the skin and core layer model (curve).

**Figure 5 polymers-15-03426-f005:**
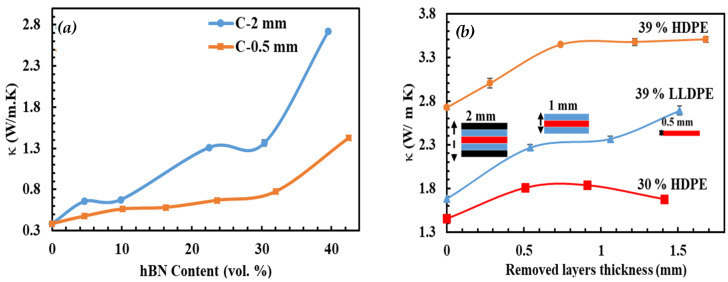
κ of C-shaped hBN/HDPE composites pressed into 2 mm and 0.5 mm thick sheets (**a**) and the κ values of C-2 mm hBN/HDPE and hBN/LLDPE composites at different removed skin layers thickness (**b**). The lines are not fitting of the data and they are only used to guide the eye.

**Figure 6 polymers-15-03426-f006:**
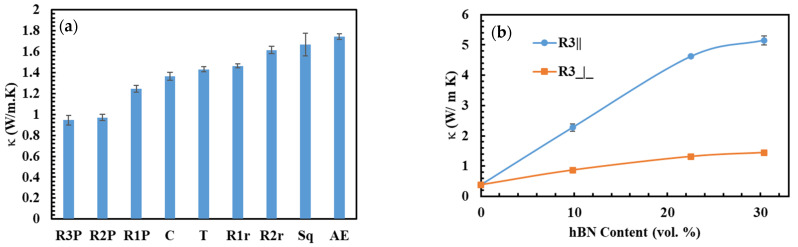
κ of (**a**) 30 vol.% hBN/HDPE composites prepared from strands of different shapes, (**b**) R3-based hBN/HPDE composites at various hBN concentrations, (**c**) 22.5 vol.% hBN/LLDPE composites of several hBN orientations, and (**d**) 30 vol.% hBN/HDPE composites prepared from strands of different shapes, pellets, and powder. vThe lines are not fitting of the data and they are only used to guide the eye.

**Figure 7 polymers-15-03426-f007:**
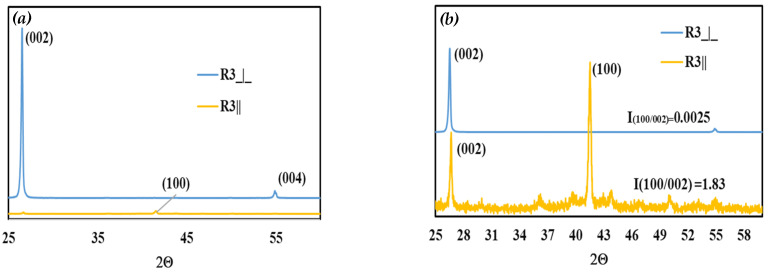
XRD spectra of R3⏊ and R3|| hBN/HDPE composites: (**a**) XRD pattern and (**b**) normalized by the (002) plane intensity.

**Figure 8 polymers-15-03426-f008:**
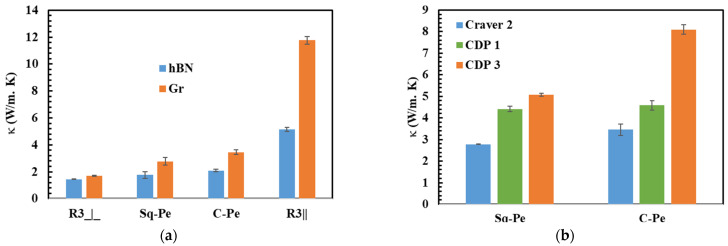
κ of 30 vol.% hBN/HDPE and Gr/HDPE composites prepared by various methods (**a**) and κ of 30 vol. % Gr/HDPE composites prepared by different compression molds. The numbers 1, 2, and 3 in (**b**) indicate the thickness of the composite sheets in ‘mm’.

## Data Availability

Raw data can be made available to the corresponding author upon request.

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
