# Peer review of "Impact of the Processing-Induced Orientation of Hexagonal Boron Nitride and Graphite on the Thermal Conductivity of Polyethylene Composites"

_polymers, 2023, doi:10.3390/polym15163426_

Round 1
Reviewer 1 Report
Dear author,
While the topic of this article is consistent with the aims & scope of Polymers, there are some suggestions for you to revise before recommending the article for publication:
1. A more in-depth revision of the summary and keywords is suggested. For example, the "к" appeared for the first time in the abstract, and it is suggested that the significance of к(thermal conductivity)is explained before it is elicited. The two key words "polyethylene" and "composite" in the key words are suggested to be merged into one keyword "polyethylene composite" and so on.
2. In 2.1.Materials, it is recommended to specify the specific specifications and brand number of each material.
3. In 2.3. Characterization, the detailed process, conditions and sample status of each experimental operation are suggested to be detailed.
4. Should the k, l, (m), (n, o), (n), (o), etc. in line 147-148 be k), l), m), n, o), n), o), etc. correspond one to one with the logo in the picture and the preceding text?
5. Is line 158 "while in the core layer, they were found to align parallel to the compression molding direction." accurate? They're more of a random arrangement, right?
6. Where is the Figure 2-d in line 217?
7. Since line 216 states that the removal method is to remove the surface layer at intervals of 0.5mm, how can the surface thickness of 0.74mm in line 220, 1.51mm in line 224 and 0.7mm in line 230 be obtained
8. It is recommended that all references be in the same format as required for polymers. For example, the authors of references 24 and 25 are formatted differently from other authors.
Reviewer 2 Report
Line 31: correct unit format
Line 44: Ghose et al.
Line 69: (b) and (c) missing in figure caption
Line 86: unit at 134 °C in next line
Line 111: c) sentence not clear. Full stop missing after "strands". Verb missing in the next sentence...
Line 147: parentheses sign after k and l missing: k) l).
l) should be in bold font.
Line 153: should be "strong dependence on thickness"
Lines 170-174: Shouldn't there be two skin layers in a sample? Shouldn't it be 2*t_skin in the formulae? If the skin thickness of 115um is taken from SEM, then the skin conductivity is ca. twice higher.
Is this a 3-parameter or actually a 2-parameter fit (skin thickness taken from SEM, k_skin, k_core fitted)?
Please give explanation and correct the results.
Line 188. Should be: "increased value of k".
Larger k may be considered improvement, but not necessarily so.
Line 191. "They were owing..." sentence not clear. What was owing what to the symmetry? Please correct.
Line 219: should be "k was becoming larger". The wording "improved" means valuation, which is not appropriate here. Please correct also in line 222 and others.
Line 225: should be "viscous"?
Line 232: deteriorated -> decreased. "deteriorated" means valuation. Not appropriate word here.
Line 239: "negative hBN orientations". What does it mean? Negatively influencing the conductivity? Please correct.
Line 257: "deteriorated" - not an appropriate word.
Line 282: improvement
Line 325: "composites were much simpler" - what does it mean? Please explain.
Line 340: improvement; line 343 idem. line 394: idem.
Line 398: correct format of unit m-1 K-1
Line 421: reference format: delete "[4]"
Line 461: remove "["
